# Microhardness Distribution of Long Magnesium Block Processed through Powder Metallurgy

**Jiaying Wang and Qizhen Li ***

School of Mechanical and Materials Engineering, Washington State University, Pullman, WA 99163, USA
* Correspondence: qizhen.li@wsu.edu

**Abstract:** Powder metallurgy is a popular method of making raw powders into specific shaped samples. However, the pressure distribution and the microhardness difference within the sample are nonnegligible and unclear when the sample is long or exceeds a specific size. In this study, the long magnesium blocks, with a ratio of about 2.8 between the sample height and the sample side length, are successfully synthesized under three uniaxial and two biaxial conditions. Then, the sample hardness values on the outer surface and the center plane are tested to study the microhardness distribution. The modified analytical expression indicates that the normal pressure exponentially decreases along the compression direction, which is consistent with the hardness distribution trend. Because higher pressure leads to a more compact arrangement of the powders, more metal bonds are formed after sintering. During the first pressing, the sidewall pressure makes the surface hardness higher. The secondary reverse compression mainly improves the bottom and core hardness due to the re-orientation and re-location of the powders. The obtained relationship between the applied pressure and the hardness distribution is instructive in predicting and improving the sample quality.

**Keywords:** magnesium powder; powder metallurgy; pressure distribution; hardness distribution

## 1. Introduction

Powder metallurgy (PM) is an excellent method of forming solid samples from raw powders by pressing and sintering [1–3]. It has a broad application field and prospects because of its simple operating processing in manufacturing [4], its high success rate [5,6], and its low energy consumption [7,8]. PM products are mainly used in the transportation market [9,10], in medical treatments [11,12], and in aerospace [10].

The ratio of height and diameter (H/Ø) is a significant limitation regarding the PM sample for industrial applications. After pressing, the basic sample size is determined. Generally, the typical causes of failure in preparing high H/Ø samples are that the powders are not compacted, or the compacts are broken. The unformed compacts cannot obtain complete samples through sintering. The sintering process has little effect on preparing high H/Ø samples by powder metallurgy. In this way, the pressing scheme is critical. The sample size obtained from different compressing parameters has been studied for a long time, and some examples in the previous research are summarized in Table 1. The applied pressure values for the different materials are between 30–1250 MPa [13–31]. Most of the height and the diameter, or side length ratios, are in the range of 0.1–1.8 [13–29]. The range of H/Ø needs to be widened to satisfy the current more comprehensive requirements. Only two examples have relatively higher ratios of about 1.875 [30] and 2.5 [31], respectively. It should be noted that their applied pressures are not the highest. Because the successful preparation conditions for powder metallurgy samples depend on many factors, such as the raw materials and their shape and size, sample shape and size, pressing scheme, mold material, and whether or not lubricant is used, the pressure cannot independently determine the H/Ø of the sample. The sample with high H/Ø may not be obtained by simply increasing the pressure. This demonstrates that multiple experiments are needed

for compacts, and higher pressures do not necessarily result in longer samples. Therefore, exploring the appropriate compression conditions for high H/Ø samples can be a valuable parameter reference in industrial production to reduce the number of experimental attempts required for long samples. In addition, the hardness of the final sample is an essential and still open topic [32–35]. The hardness difference in small and thin samples produced under high pressure is negligible [17,19,22,28,29,31,36–38]. However, the microhardness in long compacts is not constant because of severe friction and powder interaction, which must be addressed [39,40]. Such nonuniformity and variation can have a significant impact on industrial applications. So far, there is a literature gap regarding the hardness variation of the high H/Ø sample. Therefore, studying the relationship between the applied pressure and the hardness distribution is necessary.

**Table 1.** Examples of PM pressing conditions and size parameters in previous research.

| Material | Pressure (MPa) | Size (mm) | H/Ø | Reference |
|---|---|---|---|---|
| $Mg/HA/TiO_2/MgO$ | 840 | 5: Ø 12 | 0.42 | [13] |
| Mg/HA/MgO | 840 | 5: Ø 12 | 0.42 | [14] |
| AZ31/HA-Zeolite | 1000 | 20: Ø 12 | 1.67 | [15] |
| Al-Ceramic | 200; 250; 300 | 20: Ø 12 | 1.67 | [16] |
| $Al6061-Al_2O_3$ | 200–800 | 10: Ø 25 | 0.40 | [17] |
| Porous Mg Monoliths | 265 | 16: Ø 13 | 1.23 | [18] |
| Carbon Steels | 300–1250 | 9.5: Ø 8 | 1.19 | [19] |
| Al-10wt% $MoO_3$ Composite | 250; 300; 350 | 12: Ø 24 | 0.50 | [20] |
| Ti-48Al-6Nb | 300 | 2: Ø 10 | 0.20 | [21] |
| Graphene Oxide-Reinforced Al Alloy | 570 | 5: Ø 30 | 0.17 | [22] |
| Mg-3Al-1Zn Alloy | 550 | 20: Ø 82 | 0.24 | [23] |
| Nb-Ti-Al Porous Alloys | 300 | 3: Ø 32 | 0.094 | [24] |
| Ti and Polymethyl Methacrylate | 500 | 20 × 20 × 2.5 | 0.13 | [25] |
| Cu-Al-Ni | 500 | 30 × 18 × 6 | 0.33 | [26] |
| W Powder with Fisher Sub Sieve | 200–663 | 20: Ø 20 | 1.00 | [27] |
| rGO/GNS-AMC Nanocomposites | 30; 73; 220; 260; 330; 560 | 1: Ø 20 | 0.050 | [28] |
| Fe-12Mn-0.2C Alloy | 250; 500; 900 | 10: Ø 12 | 0.83 | [29] |
| Mg Powder and $NH_4HCO_3$ Powder | 265 | 30: Ø 16 | 1.88 | [30] |
| Ti-Powder and NaCl Crystals | 200 | 50: Ø 20 | 2.50 | [31] |

Note: Ø is the die diameter.

This work provides five different experimental PM conditions for successfully preparing rectangular pure magnesium (Mg) block samples with a high H/Ø of ~2.8 after many attempts. A detailed analysis of the normal pressure distribution in PM samples is performed by considering powder gravity and discussing the coefficients in the pressure expression equations. Then, thorough tests are carried out on the outer surface and the center to analyze the sample hardness distribution. The relationship between the compacting condition and the hardness is also discussed.

## 2. Materials and Methods

The raw material used consists of pure Mg powders (purity > 99.8%, ~325 mesh) in a shape with an average length of 40 μm and width of 20 μm, respectively. The image of the Mg powders, obtained by scanning electron microscopy (FEI SEM Quanta 200F, Field Emission Instruments), is shown in Figure 1a. A steel die and a punch, without a lubricant, are used to withstand high pressure. The diagram of the sample preparation process is shown in Figure 2. First, 8 g of pure Mg powder is weighted with an analytical balance at a $10^{-4}$-g precision (Adventurer™ Analytical) and then slowly filled into the die. Next, the punch bar is put into the die, and the sample is compacted by a hydraulic manual laboratory press. Then, the pressure by the press is slowly increased to a certain value and kept for a certain duration to obtain the rectangular block shaped sample. Table 2 lists five experimental plans. S1, S2, and S3 are single uniaxial press processes with a 20 min pressing duration and 143 MPa, 182 MPa, and 208 MPa of pressure, respectively. D1 and

D2 are the non-simultaneous double-direction pressing plans. After the first pressing, the die is rotated upside down, and then 104 MPa pressure is applied constantly for 5 min from the opposite direction on the moveable bottom wall. Next, the Mg block is ejected from the die slowly and sintered at 650 °C for 3 h. The heating and cooling both occur at a rate of 5 °C per minute, and the whole sintering process transpires under a protective argon gas atmosphere. The final sample, with a clean and smooth surface, is shown in Figure 1b.

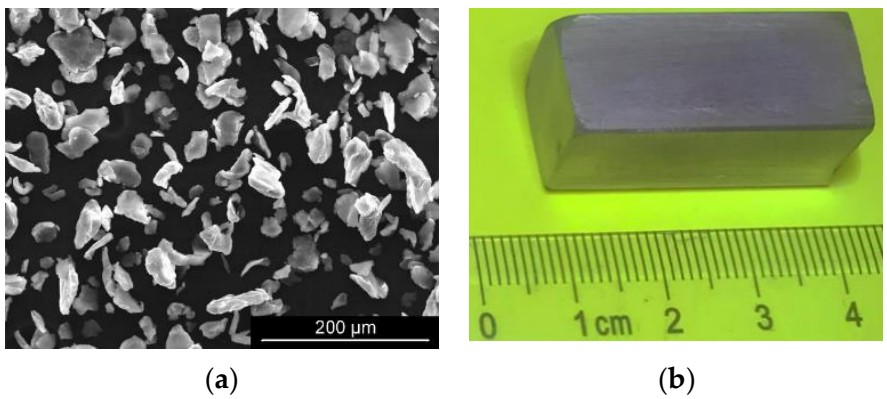

(a)  (b)

**Figure 1.** (a) Scanning electron microscopic image of pure Mg powders; (b) image of a sintered sample.

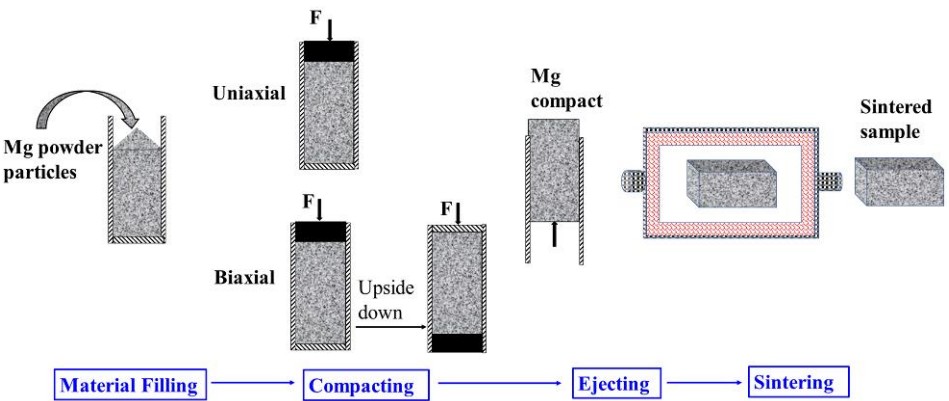

**Figure 2.** Schematic diagram of sample preparation.

**Table 2.** Process parameters followed for PM operations.

| PM Plan | First Normal Pressure | Holding Time | Second Normal Pressure | Holding Time |
|---------|----------------------|--------------|------------------------|--------------|
| S1 | 143 MPa | 20 min | – | – |
| S2 | 182 MPa | 20 min | – | – |
| S3 | 208 MPa | 20 min | – | – |
| D1 | 143 MPa | 20 min | 104 MPa | 5 min |
| D2 | 182 MPa | 20 min | 104 MPa | 5 min |

The microstructure of the final Mg sample is characterized by a ZEISS Axiocam optical microscope (OM). For the metallographic preparation, the sample is roughly and finely wet-ground with sandpapers of different grit sizes (400, 600, 800, 1200, and 2500), then polished with polish cloth on a grinding machine. After rinsing with alcohol and drying with cold air, the sample is etched using the etching solution of 1 mL alcohol, 10 mL distilled water, 1 mL 1 mol/L acetic acid, and 1 mL 4.67 wt.% picric acid. After about 25 s etching time, the etched sample is washed with alcohol and dried with cold air. The metallography photos are then obtained.

Before the hardness test, the Mg block is split into two halves in the vertical direction at half the width, as shown in Figure 3a. After polishing, the sample is cleaned with ethanol

and dried with cold air. The outer surface and center are tested using a Vickers hardness tester with a square pyramidal diamond. The hardness indentations are evenly distributed from top to bottom. There are at least ten indentations in a row along the bottom side. The distance between each row is about 2–3 mm. The actual photo after the microhardness test is shown in Figure 3c. A prismatic indentation is left at each tested position. The Vicker microhardness values are exported into a graphing program (Origin Pro Version 9.0) to produce the fitting curves and two-dimensional and three-dimensional hardness contour maps of the surface and center of the samples.

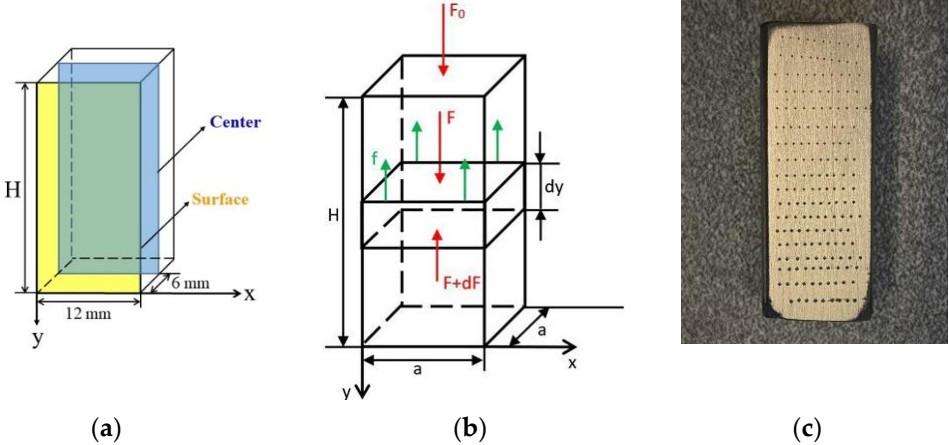

|        |        |        |
|:------:|:------:|:------:|
| (**a**) | (**b**) | (**c**) |

**Figure 3.** (**a**) Geometric diagram of the final sample and sample cut; (**b**) schematic diagram of force analysis on a slice in pressing die; (**c**) image of the sample after hardness test.

### 3. Theoretical Considerations and Experimental Results

*3.1. Theoretical Considerations*

The analysis of the normal pressure distribution in PM has a long history [41–43]. Among the existing theories, Janssen's differential slice analysis is the most widely accepted and used theoretical basis [23,44]. In Janssen's analysis, a thin slice layer of powders is selected for force analysis. The following equation is obtained according to the force–balance principle [45–50]:

$$F = (F + dF) + f \tag{1}$$

where $F$ and $(F + dF)$ represent the forces acting on the upper and lower surfaces of the thin slice, respectively; $f$ is friction between the thin powder slice and the inner mold wall, which is the main factor causing the loss of normal force $dF$, especially without lubrication [51,52]. The left side of the equal sign is the vertical downward force, and the right side is the vertical upward force, as shown in the schematic diagram in Figure 3b. This work uses the above analysis for reference by taking the Mg powder gravity $G$ into account as a correction term. The formulas are derived as follows:

$$F + G = (F + dF) + f \tag{2}$$

$$F = a^2 P \tag{3}$$

$$dF = a^2 dP \tag{4}$$

$$P_{side} = \alpha P = \frac{v}{1 - v} P \tag{5}$$

$$F_{side} = 4adyP_{side} \tag{6}$$

$$f = \mu F_{side} = \mu 4aP_{side}dy = \mu 4ady\alpha P \tag{7}$$

where $dy$ is the slice thickness in the cuboid die, and $a$ is the bottom side length. The force divided by the area on which it presses is pressure $P$. In the force analysis, the pressure

from the sidewall $P_{side}$ plays a vital role because it is directly related to the sidewall friction $f$ [23,53,54]. As above, in the Equation (5), from the experimental conclusion [23], the coefficient $\alpha$ between the normal and side pressure is related to Poisson's rate $\nu$ of the raw material [55], which is around 0.2 to 0.4 for different metal powders in different shapes [54,56]. In this work, 0.35 is taken as the $\nu$ value of pure Mg, according to the previous research results [57–59]. The calculated coefficient $\alpha$ is about 0.5385. The expression of the $F_{side}$ is obtained by multiplying the $P_{side}$ by the wall area $4ady$ of the thin slice, as shown in Equation (6). In this way, the relationship between the pressure $P$ and the friction $f$ is given in Equation (7). The powder gravity of the thin slice is expressed by the following Equation (8):

$$G = mg = v\rho g = a^2 dy\rho g \tag{8}$$

where $m$ is powder mass; $v$ is thin slice volume; $g$ is gravitational acceleration; and $\rho$ is slice density. With the substitution of Equations (3), (4), (7), and (8) into Equation (2), Equation (9) is obtained. Then, the differential expression of normal pressure $dP$ is derived according to the following Equations (10)–(13):

$$a^2 P + a^2 \rho g dy = a^2 (P + dP) + 4a\mu\alpha P dy \tag{9}$$

$$a^2 P - \left[ a^2 (P + dP) \right] = 4a\mu\alpha P dy - a^2 \rho g dy \tag{10}$$

$$-a^2 dP = 4a\mu\alpha P dy - a^2 \rho g dy \tag{11}$$

$$\left( \frac{4\mu\alpha P}{a} - \rho g \right) dy = -dP \tag{12}$$

$$-dy = \frac{1}{\frac{4\mu\alpha P}{a} - \rho g} dP \tag{13}$$

The friction coefficient $\mu$ is another critical parameter in the PM process [55,60,61]. It is determined by various physical parameters and material properties [62–65]. For the metal powder and a rigid metal mold, the relative density $\rho_r$ is the most critical determinant for $\mu$, which is also revealed in the following equation [23].

$$\mu = 0.077\rho_r^{-1.87} \tag{14}$$

After all the parameters are determined, the final expression is obtained by integrating Equation (13). Where $P_0$ is the applied normal pressure, the thin slice location corresponds to the distance $y$ from the top of the sample ($0 < y < H$). $H$ is the height of the final green compact [23,66,67]. The resulting exponential function of the pressure distribution shows that the pressure decreases from top to bottom with the increase in $y$. Moreover, the downward trend becomes smaller with the increasing distance.

$$-\int_0^y dy = \int_{P_0}^P \frac{1}{\frac{4\mu\alpha P}{a} - \rho g} dP \tag{15}$$

$$-y = \int_{P_0}^P \frac{1}{\frac{4\mu\alpha P}{a} - \rho g} dP = \frac{a \left[ ln\left( -\rho g + \frac{4\mu\alpha P}{a} \right) - ln\left( -\rho g + \frac{4\mu\alpha P_0}{a} \right) \right]}{4\mu\alpha} \tag{16}$$

$$-\frac{4\mu\alpha}{a} y = ln\left( -\rho g + \frac{4\mu\alpha P}{a} \right) - ln\left( -\rho g + \frac{4\mu\alpha P_0}{a} \right) \tag{17}$$

$$e^{\left( -\frac{4\mu\alpha y}{a} \right)} = e^{ln\left( -\rho g + \frac{4\mu\alpha P}{a} \right) - ln\left( -\rho g + \frac{4\mu\alpha P_0}{a} \right)} = \frac{-\rho g + \frac{4\mu\alpha P}{a}}{-\rho g + \frac{4\mu\alpha P_0}{a}} \tag{18}$$

$$e^{\left(-\frac{4\mu\alpha y}{a}\right)} = \frac{\left(-\rho g + \frac{4\mu\alpha P}{a}\right) \times \frac{a}{4\mu\alpha}}{\left(-\rho g + \frac{4\mu\alpha P_0}{a}\right) \times \frac{a}{4\mu\alpha}} = \frac{P - \frac{\rho g a}{4\mu\alpha}}{P_0 - \frac{\rho g a}{4\mu\alpha}} \qquad (19)$$

$$\left(P_0 - \frac{\rho g a}{4\mu\alpha}\right) \times e^{\left(-\frac{4\mu\alpha y}{a}\right)} = P - \frac{\rho g a}{4\mu\alpha} \qquad (20)$$

$$P = \left(P_0 - \frac{\rho g a}{4\mu\alpha}\right) \times e^{\left(-\frac{4\mu\alpha y}{a}\right)} + \frac{g a}{4\mu\alpha} \qquad (21)$$

### 3.2. Sample Microstructures

The microstructures of the uniaxial and biaxial samples prepared under different conditions are shown in Figures 4 and 5. The yellow regions show the Mg powders, and the black areas are the junctions. The powder arrangements inside all the samples are very dense. It can be observed that the shape and the size of the powders in the S1 sample are the most uniform. Most of the powders retain their initial shape and interlock tightly together. In the S2 sample, the shape of the powders becomes slender due to compression, and the contact area between the powders is greater than in S1. Furthermore, more deformed areas are obtained in the S3 sample with increasing pressure. After the second opposite-direction pressing, the microstructure of the D1 sample significantly changes compared with that of the S1 sample. It can be seen from Figure 5 that the second pressing in D1 results in the orientation change and some deformation areas in the powders. However, D2 shows no noticeable change compared with S2, since the reverse pressure is not very high, and the D2 sample has experienced powder deformation before reverse pressing.

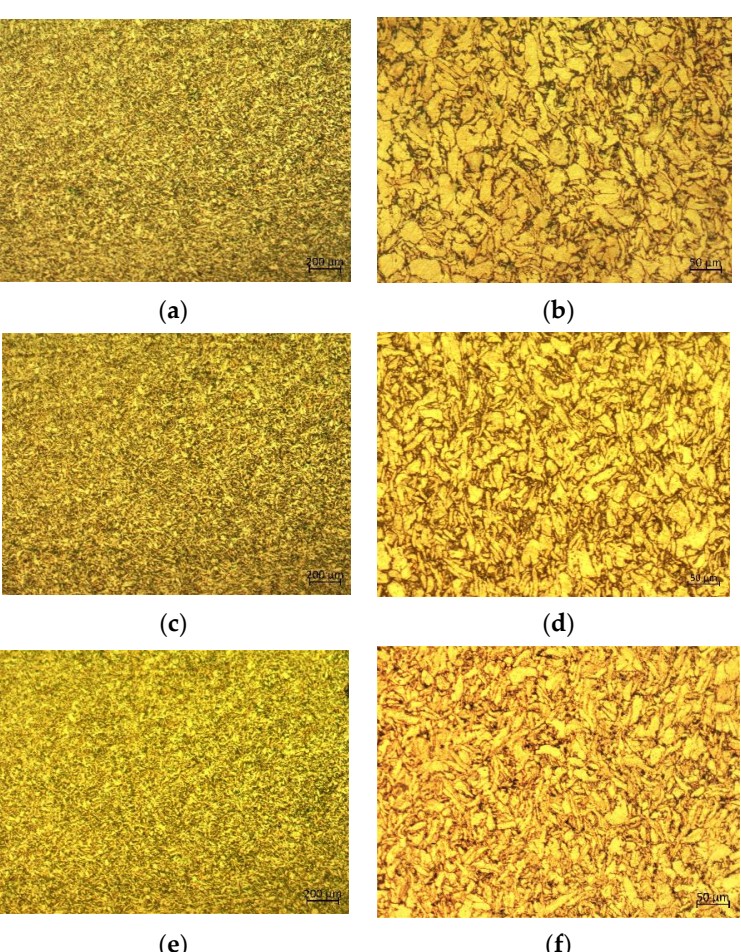

(**a**)　　　　　　　　　　　　　(**b**)

(**c**)　　　　　　　　　　　　　(**d**)

(**e**)　　　　　　　　　　　　　(**f**)

**Figure 4.** Microstructures of uniaxial PM Mg under (**a**,**b**) S1, (**c**,**d**) S2, and (**e**,**f**) S3 conditions.

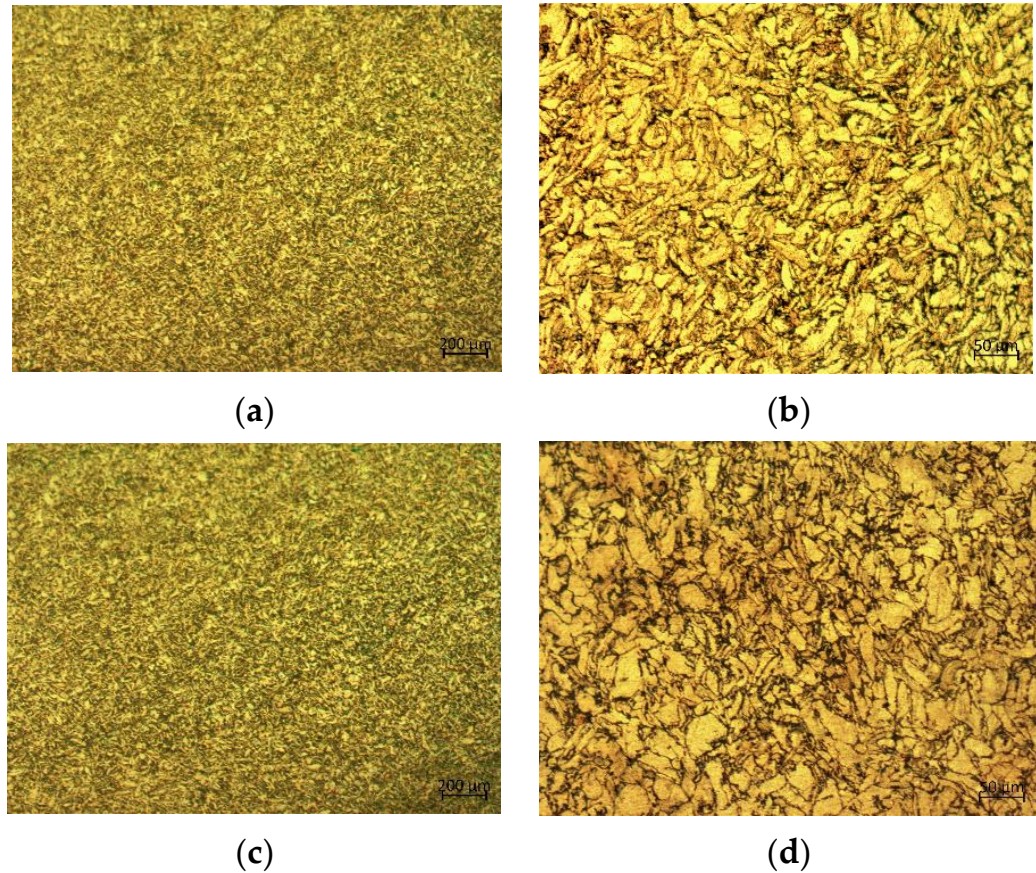

**Figure 5.** Microstructures of biaxial PM Mg under (**a**,**b**) D1 and (**c**,**d**) D2 conditions.

### 3.3. Hardness Distribution Test Results and Discussion

The sample hardness of the surface and center is tested and analyzed. Table 3 shows the average extremum values of each kind of sample. The hardness distributions of the three types of uniaxial samples are similar. Sample S1−1 represents uniaxial samples to show the raw microhardness test data, as shown in Figure 6. Repeated tests cannot be performed at the same point to obtain the bars because it is damaged after the test. Two biaxial pressing schemes exhibit similar trends in microhardness distribution. Sample D1−1 in Figure 7 is representative of the biaxial samples. After the fitting curve is obtained, the sample length is unified as 34 mm for further discussion and comparison.

**Table 3.** Average hardness value at different positions in the samples.

| PM Plan | Surface Hardness (HV) | | | Center Hardness (HV) | | |
|---|---|---|---|---|---|---|
| | **Top** | **Middle** | **Bottom** | **Top** | **Middle** | **Bottom** |
| S1 | $30.4 \pm 2.6$ | $12.9 \pm 0.5$ | $8.9 \pm 1.5$ | $29.9 \pm 0.7$ | $12.5 \pm 1.9$ | $11.7 \pm 2.0$ |
| S2 | $34.0 \pm 2.8$ | $15.7 \pm 1.0$ | $9.2 \pm 1.5$ | $29.1 \pm 0.7$ | $14.9 \pm 1.9$ | $10.4 \pm 2.3$ |
| S3 | $41.9 \pm 3.1$ | $16.2 \pm 2.1$ | $10.5 \pm 0.6$ | $37.4 \pm 2.0$ | $15.4 \pm 0.5$ | $11.2 \pm 1.1$ |
| D1 | $24.2 \pm 1.7$ | $12.4 \pm 3.2$ | $16.7 \pm 4.5$ | $29.4 \pm 1.9$ | $13.7 \pm 1.5$ | $18.7 \pm 3.3$ |
| D2 | $32.0 \pm 4.1$ | $14.9 \pm 3.2$ | $21.4 \pm 8.6$ | $31.5 \pm 1.6$ | $15.5 \pm 0.7$ | $21.2 \pm 1.7$ |

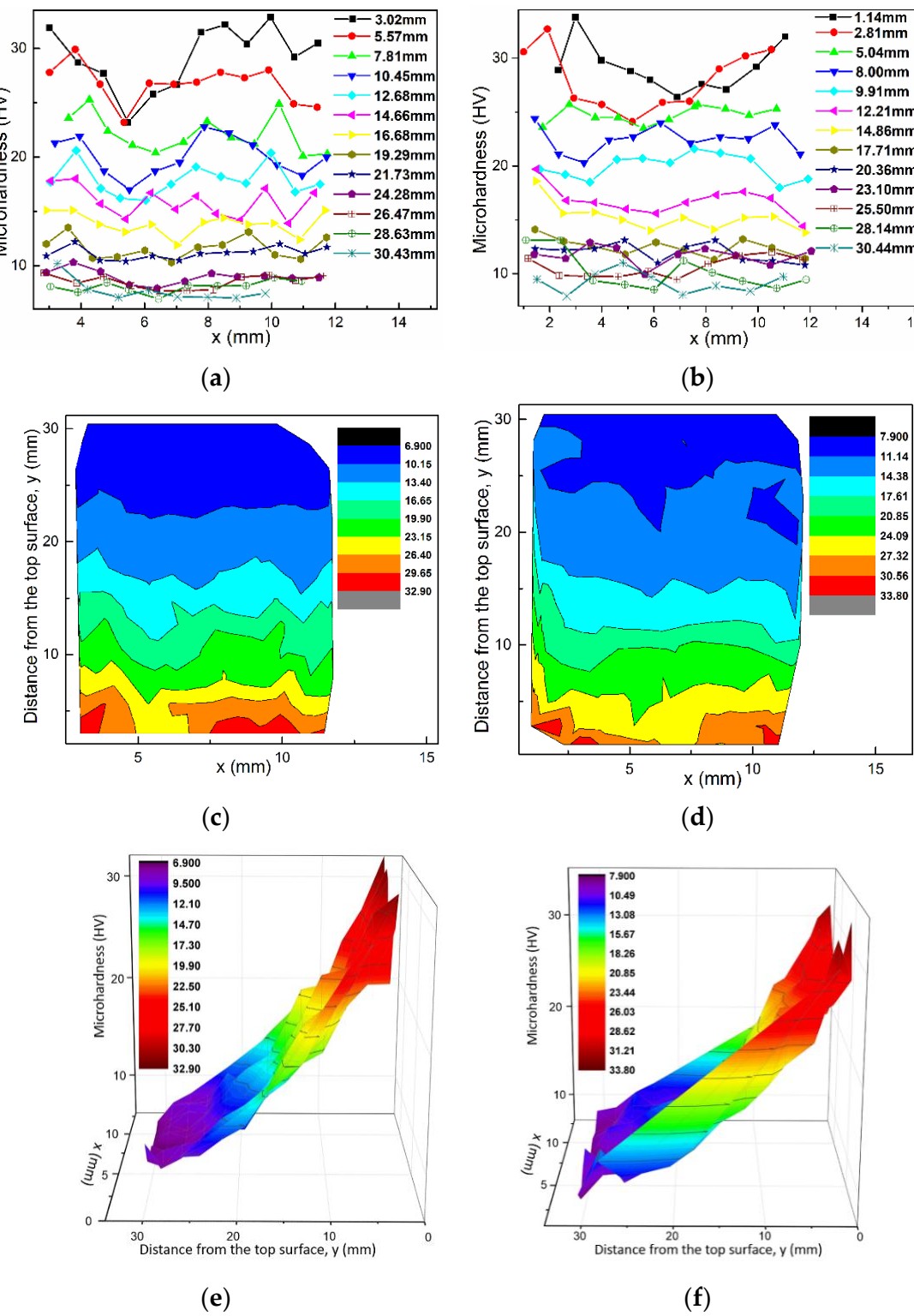

**Figure 6.** Hardness test results of S1−1 sample on the (**a**) surface and (**b**) center; hardness contour plot of S1−1 sample (**c**) surface and (**d**) center; hardness 3D plot of S1−1 sample (**e**) surface and (**f**) center.

In the uniaxial samples, some regularities regarding the hardness distribution are obtained according to the original microhardness data, the fitting curve, and the calculated hardness at both ends. The hardness decreases with an increasing distance from the top surface. The exponential function shows the best fitting expression by comparing the fitting

results of various functions of Origin Pro Version 9.0 software. The function of each uniaxial sample is shown in Table 4. Maximum and minimum hardness values can be estimated from the $M_f$ fitting curve equation. They are indicated and calculated by $M_f$ ($y = 0$) and $M_f$ ($y = 34$). During compaction, Mg powders undergo a series of rearrangements, elastic deformations, and plastic deformations [68]. Then, Mg powders are bonded by atomic diffusion during the sintering process. At the same time, new bonds between the metallic powders are formed [68]. The higher the applied pressure, the more compact powder particles are obtained and the more metallic bonds are created after sintering to resist the deformation, as reflected in the higher values in the hardness test. Therefore, the hardness is a function of compacting pressure. The exponential function is easy to understand because the pressure distribution can also be expressed by an exponential equation.

**Table 4.** Uniaxial sample surface, center hardness fitting curve equation, and calculated microhardness on the top and bottom ends.

| Sample | Surface Hardness Fitting Curve Equation | $M_f$ ($y = 0$) | $M_f$ ($y = 34$) | Center Hardness Fitting Curve Equation | $M_f$ ($y = 0$) | $M_f$ ($y = 34$) |
|---|---|---|---|---|---|---|
| S1−1 | $33.909e^{-0.045y}$ | 33.91 | 7.34 | $29.981e^{-0.030y}$ | 29.98 | 10.81 |
| S1−2 | $30.108e^{-0.046y}$ | 30.11 | 6.30 | $27.309e^{-0.033y}$ | 27.31 | 8.89 |
| S1−3 | $33.772e^{-0.051y}$ | 33.77 | 5.96 | $29.641e^{-0.041y}$ | 29.64 | 7.35 |
| S2−1 | $40.375e^{-0.053y}$ | 40.38 | 6.66 | $28.370e^{-0.039y}$ | 28.37 | 7.53 |
| S2−2 | $40.548e^{-0.055y}$ | 40.55 | 6.25 | $31.323e^{-0.047y}$ | 31.32 | 6.34 |
| S2−3 | $29.807e^{-0.042y}$ | 29.81 | 7.15 | $32.365e^{-0.037y}$ | 32.37 | 9.20 |
| S3−1 | $48.468e^{-0.063y}$ | 48.47 | 5.69 | $42.706e^{-0.060y}$ | 42.71 | 5.55 |

In uniaxial pressing, high-hardness samples can be obtained by increasing the pressure. However, the relationship between the applied compacting pressure and the overall hardness is not positively linear. It is shown in Figure 8 that the fitting curves of S1 and S2 are near each other. The S3 curve is above them. This means that the increase in hardness is not significant when the applied pressure is improved from 143 MPa to 182 MPa. Because the room available for the rearrangement and reorientation of the powders is limited when the compact is close to the dense space saturation, the powders are not remarkably better packed by increasing the pressure to 39 MPa. In this case, the plastic deformation of the powders is the main factor in improving the hardness. This can only be achieved by further increasing the pressure. In addition, when the applied pressure increases to 208 MPa, the end of the S3 hardness–fitting curve overlaps those of the S1 and S2 curves. Since the applied force is transferred from the top to the bottom through the push between the powders, after the powders are close to the saturation of rearrangement and elastic deformation, this transfer is more difficult [69,70]. This is because the interstice required for the further movement and rotation of the powders is insufficient [71,72]. As a result, in the S3 sample, the noticeable hardness improvement mainly appears in the upper part of the sample. After the powders at the top initially reaches the saturated powder arrangement, the hardness improvement in the S3 upper part is primarily attributed to the high densification from the severe plastic deformation of the powders. However, this causes a high-stress concentration between the powders, resulting in a low success rate of sample preparation [73,74]. The shapes of the average hardness–distance curve on the outer surface and the center are the same. However, the hardness of the center is lower. This is because the side wall pressure perpendicular to the axis also promotes the dense arrangement and deformation of the powders. The side pressure also acts from the outer surface to the core through the transmission between powders and is lost due to the friction between powders. Therefore, raw materials close to the sample surface are better densified, so the hardness is higher than that of the center.

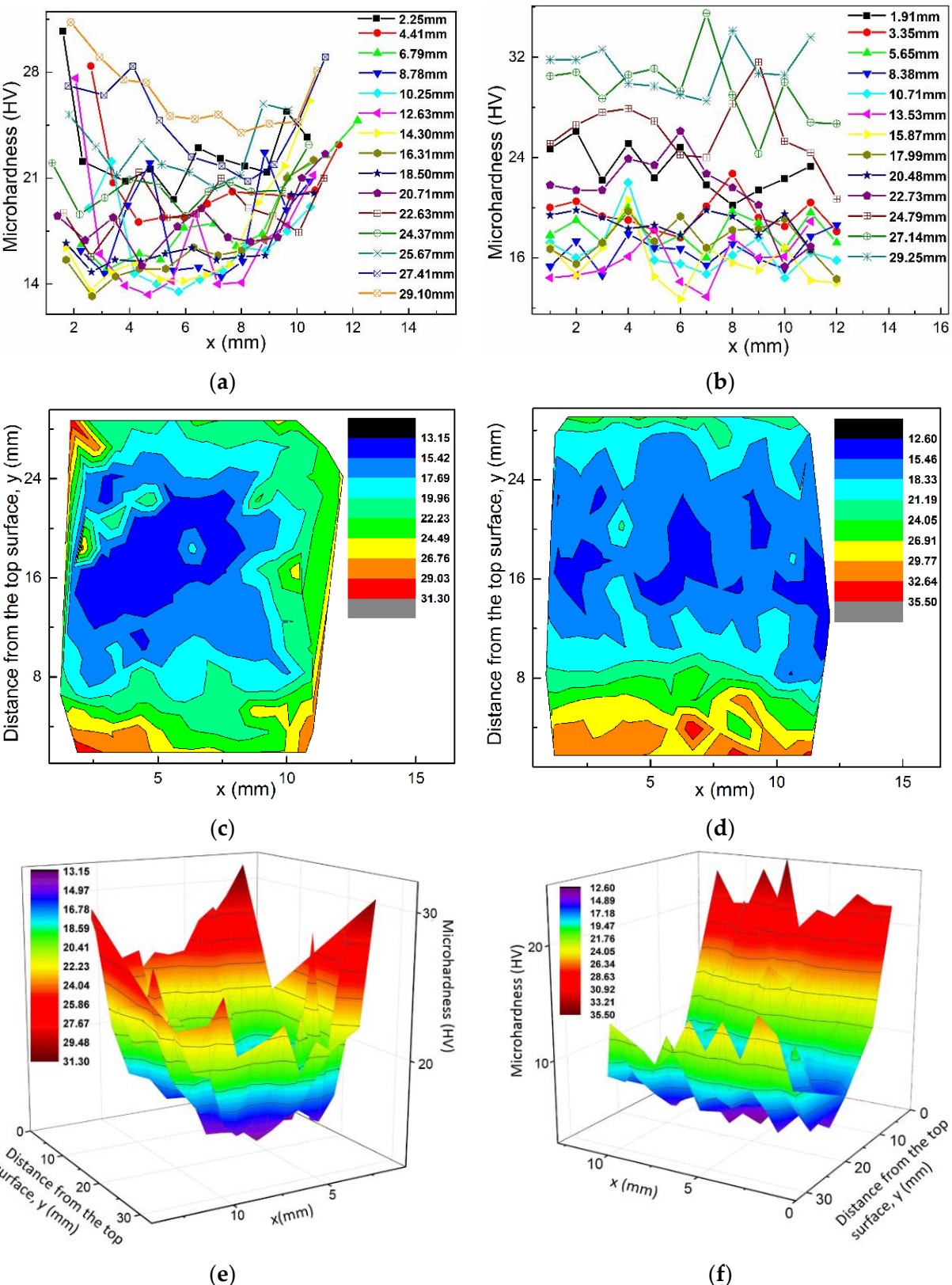

**Figure 7.** Hardness results of D1−1 sample (**a**) surface and (**b**) center; hardness contour plot of D1−1 sample (**c**) surface and (**d**) center; hardness 3D plot of D1−1 sample (**e**) surface and (**f**) center.

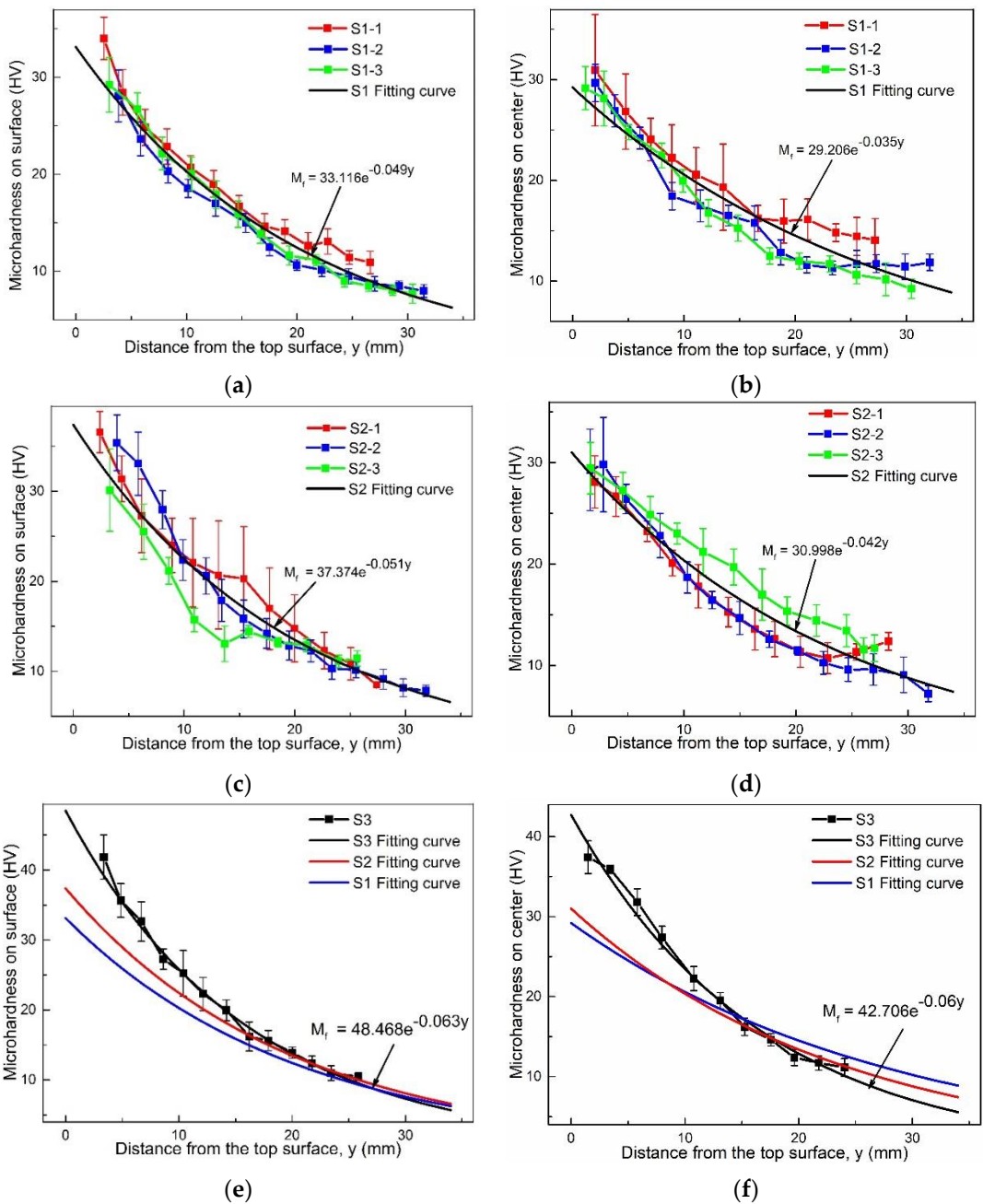

**Figure 8.** S1−1, S1−2, and S1−3 sample microhardness scatter plots and S1 microhardness fitting curves on the (**a**) surface and (**b**) center; S2−1, S2−2, and S2−3 sample microhardness scatters and S2 microhardness fitting curves on (**c**) surface and (**d**) center; S3 sample microhardness scatters and fitting curves, S1 hardness fitting curve, and S2 hardness fitting curve on the (**e**) surface and (**f**) center.

Compared with the uniaxial samples, the D1 and D2 hardness distribution curves have different shapes and value ranges, as shown in Figure 9. From top to bottom, the hardness decreases first and then increases. The minimum hardness value appears at the distance $h$ from the top. The functions in Table 5 can describe the average hardness fitting curves of the surface and the center:

$$\begin{cases} M_f = Ae^{By}(0 \leq y \leq h) \\ M_f{'} = A{'}e^{B{'}y} \ (h \leq y \leq 34) \end{cases} \tag{22}$$

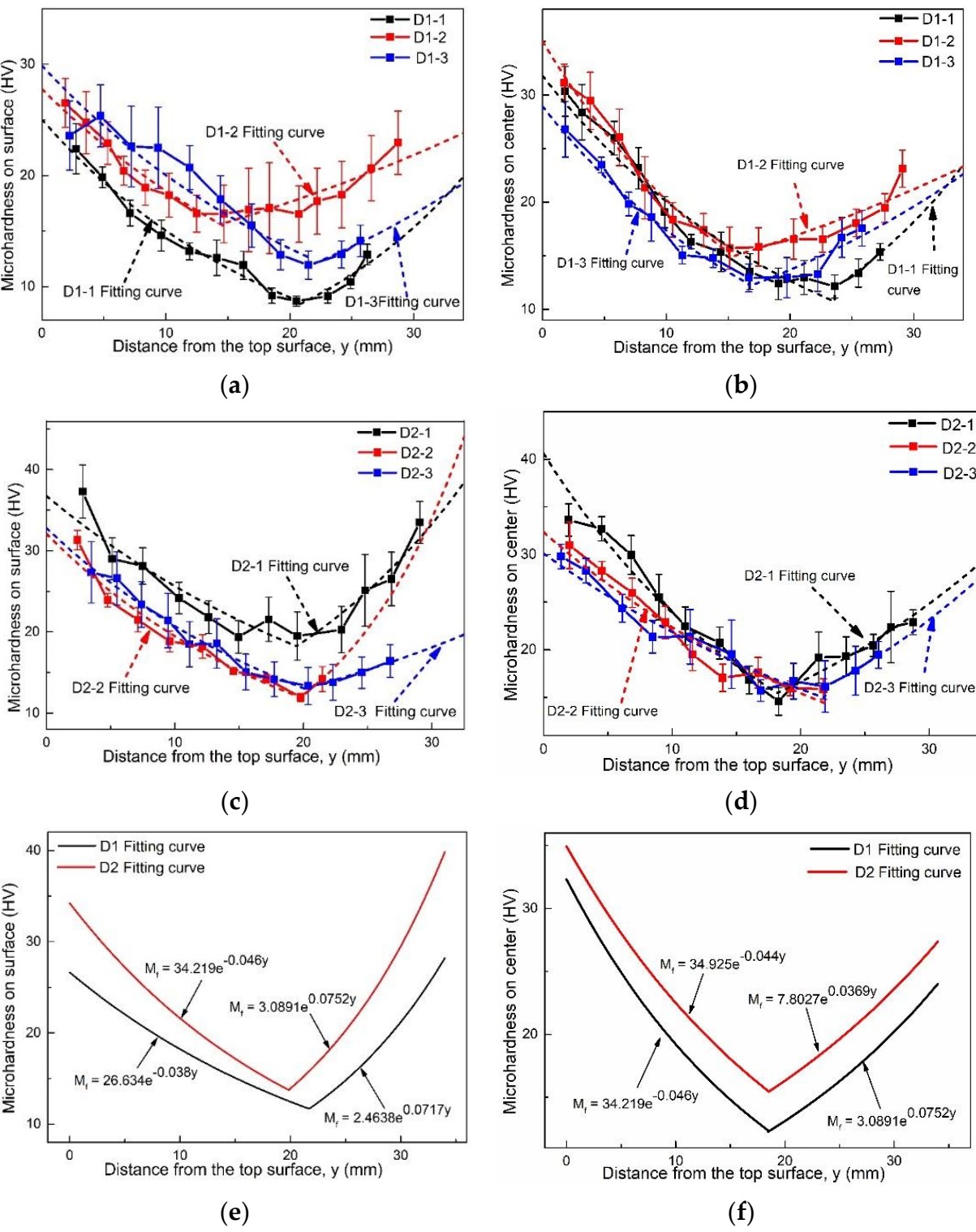

**Figure 9.** D1−1, D1−2, and D1−3 sample microhardness scatters and fitting curves on (**a**) surface and (**b**) center; D2−1, D2−2, and D2−3 sample microhardness scatters and fitting curves on (**c**) surface and (**d**) center; D1 and D2 hardness fitting curves on (**e**) surface and (**f**) center.

**Table 5.** Biaxial sample surface and center hardness fitting curve equations.

| Sample | Surface Hardness Fitting Curve Equation | Center Hardness Fitting Curve Equation |
|---|---|---|
| D1−1 ($h$ = 20.53) | $\begin{cases} 24.992e^{-0.051y}\,(0 \leq y \leq h) \\ 2.1687e^{0.065y}\,(h \leq y \leq 34) \end{cases}$ | $\begin{cases} 31.779e^{-0.046y}\,(0 \leq y \leq h) \\ 2.638e^{0.0643y}\,(h \leq y \leq 34) \end{cases}$ |
| D1−2 ($h$ = 14.67) | $\begin{cases} 27.774e^{-0.040y}\,(0 \leq y \leq h) \\ 11.429e^{0.0216y}\,(h \leq y \leq 34) \end{cases}$ | $\begin{cases} 34.998e^{-0.055y}\,(0 \leq y \leq h) \\ 10.351e^{0.0239y}\,(h \leq y \leq 34) \end{cases}$ |
| D1−3 ($h$ = 21.5) | $\begin{cases} 29.876e^{-0.040y}\,(0 \leq y \leq h) \\ 5.1271e^{0.039y}\,(h \leq y \leq 34) \end{cases}$ | $\begin{cases} 28.945e^{-0.051y}\,(0 \leq y \leq h) \\ 6.675e^{0.0359y}\,(h \leq y \leq 34) \end{cases}$ |
| D2−1 ($h$ = 19.52) | $\begin{cases} 36.741e^{-0.036y}\,(0 \leq y \leq h) \\ 6.003e^{0.0571y}\,(h \leq y \leq 34) \end{cases}$ | $\begin{cases} 40.595e^{-0.053y}\,(0 \leq y \leq h) \\ 7.4551e^{0.0401y}\,(h \leq y \leq 34) \end{cases}$ |
| D2−2 ($h$ = 19.78) | $\begin{cases} 32.09e^{-0.050y}\,(0 \leq y \leq h) \\ 7.5747e^{0.0528y}\,(h \leq y \leq 34) \end{cases}$ | $\begin{cases} 32.353e^{-0.037y}\,(0 \leq y \leq h) \\ -----\,(h \leq y \leq 34) \end{cases}$ |
| D2−3 ($h$ = 20.36) | $\begin{cases} 32.794e^{-0.046y}\,(0 \leq y \leq h) \\ 6.7055e^{0.0331y}\,(h \leq y \leq 34) \end{cases}$ | $\begin{cases} 30.098e^{-0.032y}\,(0 \leq y \leq h) \\ 6.0142e^{0.0449y}\,(h \leq y \leq 34) \end{cases}$ |

The hardness decreases following an exponential function along $M_f$ and $M_f\prime$ with the two pressing directions, just as $B$ and $B'$ have opposite signs. D2−2 was fractured, and the distance from the top surface to the fracture is about 23 mm. The distance from the fracture to the bottom of D2−2 is about 8 mm. Although the lower part of the hardness curve is not shown, the top 23 mm on this sample still conforms to the exponential function.

Both the surface and center hardness curves of D2 are above those of D1. The values of $A$ and $A'$ in D2 are higher than those in D1. This result mainly depends on the first compression, since the reverse pressing conditions of the two schemes are the same. The higher the first pressure, the higher the hardness. It is the same as the hardness trend of the uniaxial samples. Although the second pressing uses low pressure and has a short loading time, the hardness of the bottom is significantly improved. Besides, the hardness value of the center exceeds that of the outer surface in the biaxial samples, which is opposite to the values of the uniaxial samples because in the opposite second compression, the pressure is applied on the Mg compact. It is not applied to the Mg powders as it is in the first compression. During the reverse pressing, the powders are re-orientated and re-located [75,76]. Some powders are forced to move into the pores after the first pressing, which are located mainly at the bottom and core of the sample. Moreover, ellipse-shaped powders could also act as a critical factor for this increase because they provide a better interlocking of the particles [72,74]. Therefore, when a second pressure is loaded to the compact, the hardness of the sample is significantly improved, especially in the bottom and the core.

## 4. Conclusions

In this work, long Mg blocks are obtained from the three uniaxial and two secondary-reverse PM plans. Overall, the uniaxial 143 MPa pressure condition is the best one to obtain a sample with a high H/Ø of about 2.8. The relatively high normal pressure and the second opposite compression may lead to a low success rate of the sample preparation, due to the high-stress concentration between the powders.

The hardness distributions show that the higher the pressure is, the more compact the powders are, and there are more newly formed metallic bonds and a higher hardness in the final samples. The hardness decreases exponentially along the compression direction from top to bottom. Moreover, the relationship between normal pressure and distance along the applied pressure direction agrees well with trend of hardness distribution, considering powder gravity in the force analysis. When the powders at the top first reach the saturated particle arrangement, the pressure transmission to the bottom is hindered. The improvement in the hardness occurs mainly in the upper part of the samples as the applied pressure increases because of the plastic deformation of the powders. The surface hardness of the uniaxial samples is higher than the center hardness because the pressure

from the sidewall causes a denser arrangements of the powders. After the second pressing in opposite directions, the hardness of the sample decreases first and then increases, from top to bottom, in the biaxial sample. The hardness–distance curves can be divided into two parts, each of which can be fitted by an exponential equation. The hardness of the lower part and the center of the samples is improved after the second compression in the biaxial samples.

**Author Contributions:** Conceptualization and methodology, J.W. and Q.L.; validation, J.W.; formal analysis, J.W.; writing—original draft preparation, J.W.; writing—review and editing, J.W. and Q.L.; supervision, Q.L.; project administration, Q.L.; funding acquisition, Q.L. All authors have read and agreed to the published version of the manuscript.

**Funding:** This research was funded by US Department of Energy, award number DESC0016333.

**Informed Consent Statement:** Not applicable.

**Data Availability Statement:** The data will be provided by the corresponding author upon request.

**Conflicts of Interest:** The authors declare no conflict of interest.

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
