# Peer review of "Microhardness Distribution of Long Magnesium Block Processed through Powder Metallurgy"

_jmmp, doi:10.3390/jmmp7010005_

Round 1

Reviewer 1 Report

The authors have to make some small changes:
- The authors must justify the way in which they chose the values of the compaction pressures used in the determinations.
- The symbol for diameter, in the International System of Units, ISO, is Ø not

φ (Table 1) .
- The authors specify that they made samples D1 and D2 by pressing in two directions, but in figure 2 it appears pressing in one direction (along y) and uniaxial (from top to bottom), not biaxially, how do they write???? Figure 2 should be redone for clarification.

Author Response

Dear Editors and Reviewers:

Thank you very much for your careful review and constructive suggestions about our manuscript "Microhardness Distribution of Long Magnesium Block Processed through Powder Metallurgy" (jmmp-2098590). The comments are all valuable and very helpful for revising and improving our paper, as well as the importance of guiding significance to our research. We have studied the comments carefully and have made corrections which we hope meet with approval. Revised portions are tracked in the manuscript. The primary corrections and explanations in the manuscript and the responses to the reviewer's comments are as follows (as appearing in sequence):

  1. Considering the suggestion, we have added a brief description of the minor effects of the sintering process on preparing high H/Ø samples by powder metallurgy in the Introduction. After pressing, the sample size is basically determined. The size will not change significantly after sintering. The higher H/Ø value sample is unlikely to be prepared through optimization of the sintering process. Therefore, this paper mainly discusses pressing instead of sintering. Thank you for pointing this out.
  2. In the previous manuscript, we did not clearly explain the influence mechanism of pressing pressure on preparing a high H/Ø ratio by powder metallurgy. Sorry for this confusion. More detailed explanations have been added in the Introduction. It can be seen from the literature review and Table 1 that the longest sample was not prepared under the highest pressing pressure. Since H/Ø is not proportional to the applied pressure, the successful high H/Ø preparation schemes are instructive for future sample preparation. They can be used as a reference to reduce the number of sample preparation attempts.
  3. Sorry for our carelessness. The diameter symbol has been changed to the International System of Units, Ø.
  4. The brand and unit of the analytical balance measurement and more details about the pressing have been added in Section 2, Materials and Methods. We are very sorry for our carelessness. In addition, it needs to be stated that the hydraulic manual laboratory press machine is old, and its brand is unknown.
  5. As the reviewer thought, it is true that the bottom wall in the second compaction was missing. It has been added in the revised Figure 2.
  6. We are very sorry for our negligence that the fonts in some figures were too small. The fonts in Figure 2,  Figure 6. (a), Figure 6. (b), Figure 8. (a), and Figure 8. (b) have been enlarged. High-definition versions of all the pictures have also been provided.
  7. The values of the compaction pressures used after many experimental schemes have been carried out. Different applied pressure, loading times, and powder masses were tried to find suitable pressing conditions. After eliminating some inappropriate schemes, the appropriate ones were found and verified repeatedly.
  8. After carefully examining the comments, we must admit that only the surface and the center were tested. The reason is that the bottom edge of the sample is a square. And the sample was cut vertically at half the width before testing. On the surface and central plane of the sample, the hardness of the sample was tested from top to bottom, and the distance between each row of test points was about 2-3 mm. There were at least ten dents in a row along the bottom edge of the sample. The image of the HV tested sample has been included in the revised manuscript as the new Figure 3. (c) to facilitate an understanding of the test point locations. The data in each row reflects the hardness change in the horizontal direction. Each column represents a change in the vertical direction. The hardness change in the horizontal direction can be seen more clearly from the 2D diagrams [Figure 6. (c), Figure 6. (d), Figure 8. (c), and Figure 8. (d)]. The 3D diagrams [Figure 6. (e), Figure 6. (f), Figure 8. (e), and Figure 8. (f)] can reflect the hardness change trend of samples from top to bottom and from outside to inside.
  9. The software (Origin Pro Version 9.0) was used for the mathematical processing of the experimental results and the diagram development has been added.
  1. Equation (1) comes from the Janssen analysis. As the reviewer thought, removing the parentheses leads to . It can be understood as the loss of applied pressure caused by friction. The revised details have been added.
  2. The gravity of the thin slice is indeed negligible. We added gravity into consideration to make formula derivation more accurate because the gravities of material powder differ in a large range for different materials, such as ceramic and metals.
  3. Considering these kind suggestions, we have rewritten Section 3.1 in the revised manuscript to explain the force analysis in detail. The explanations of the symbols have been inserted after the equations, and the word “where” has been used. Sorry for the inconvenience in your reading. We also have shown the calculation process step by step for a better understanding and clarity of the pressure distribution.
  4. As suggested by the reviewer, Figure 4 and Figure 5 have been referred into the revised manuscript to reveal the microstructures of the PM samples. The metallographic preparation and explanations of the microstructures have been added.
  5. There was no error bar at each hardness test point. Because the indenter is a square pyramidal diamond, the point after the test was damaged, and repeated tests could not be performed. Each kind of sample had three tested ones to ensure the reliability of the data. Except for S3, there was only one sample due to the low successful preparation rate.
  6. For hardness scatter fitting, several functions were applied, such as the exponential function, linear function, logarithmic function, polynomial function, and power multiplication function. The exponential function fitted the experimental results best. Another justification for using exponential functions is that the applied pressure distribution is also expressed by an exponential equation, as discussed in Section 3.1
  7. We did not express our meaning correctly about the metallic bonds in the previous manuscript. Sorry for this confusion. As the reviewer thought, the hardness of the PM products is obtained from the sintering process applied after compaction. In the revised version, the “more metallic bonds are created to resist the deformation” has been corrected to “more metallic bonds are created after sintering to resist the deformation.”
  8. We inferred that the greater the density of the sintered sample, the higher the hardness. For the high H/Ø sample in this experiment, the cross-section of the sample is small, and the height is very high. Therefore, the density difference would be obvious in the vertical direction. The density difference between the sample surface and core is minimal. Besides, the hardness test points in this experiment are dense, which is impossible for a density test. The density change test and mechanism analysis of samples from top to bottom are in progress and will be discussed in another paper.
  9. All typos in References have been revised and removed.
  10. The language in our revised manuscript was revised thoroughly, and we hope it meets the final publication standard. Thank you very much for your precious comments.

A point-by-point response to the reviewer’s comments is provided. Please see the attachment.

We appreciate the Editors/Reviewers' work earnestly and hope the correction will meet with approval. Correspondence should be directed to Professor Qizhen Li (qizhen.li@wsu.edu, School of Mechanical and Materials Engineering, Washington State University, Pullman, 99164, USA). Special thanks to you for your good comments and suggestions.

Very sincerely yours,

Jiaying Wang

Qizhen Li

Reviewer 2 Report

The paper discusses an important topic of microhardness distribution of long magnesium block prepared by powder metallurgy (PM). Appropriate compression conditions are the key to prepare long samples (high H/D) in this paper. This work provides five different experimental PM conditions for successfully preparing rectangular pure magnesium block samples with a high H/D of about 2.8. With overall consideration, the uniaxial 143 MPa pressure condition is the best one to obtain the sample with a high H/D of about 2.8. This is an interesting and valuable work that can provide a valuable reference for researchers in the PM of long size components. However, some suggestions could be made for authors' consideration.

1) The hardness distribution along the longitudinal direction can be observed for the cut sample shown in Fig. 3, while only two groups of parameters can be observed for the horizontal hardness. What is the horizontal hardness distribution law? It is recommended to add horizontal cutting samples and hardness distribution.

2) Is the high hardness of the sample surface only related to the density? How does the density of the sample change from the surface to the center? The higher the pressing pressure, the higher the density of the sample? What are the actual micrographs of the samples under different pressures?

3) Fig. 4 shows the microhardness at different positions of the sample, but there is no error bar. Each point is measured only once?

4) Fig. 4 shows only the hardness results. What is the actual micrograph of the sample after microhardness pressurization?

5) The author points out that the higher the pressing pressure is, the higher the density of the sample is, but the longest sample is not prepared under the highest pressing pressure. What is the relationship between the pressing pressure and the H/D ratio of the sample? What is the influence mechanism of pressing pressure on the preparation of high H/D ratio by powder metallurgy? Please discuss further.

6) The whole article lacks the microstructure diagram, please supplement.

7) Effect of sintering process on preparation of high H/D samples by powder metallurgy? Can higher H/D value samples be prepared through optimization of sintering process?

Author Response

(The authors gave the same response as above.)

Reviewer 3 Report

1. The subject addressed in the paper is of current interest and consistent with the journal's profile.

2. The authors used modern research equipment.

3. The authors also considered the force of gravity in ensuring the pressure exerted on the metal powder. Could the value of the gravitational force be highlighted within the compressive force?

4. On page 2, line 65, an analytical balance accuracy of 10-4 is indicated, but without mentioning the units of measurement used to highlight the accuracy.

5. In figure 2, in the third image from the bottom row, the bottom wall, which would allow the second compaction, seems to be missing.

6. Relationship (1) is not clear enough. Removing the parentheses seems to lead to dF+f=0.

7. When an equation is introduced, an immediate explanation of the symbols is required. Such a convention is not respected in the case of equations (2)-(9), etc.

8. I think some explanation of what equations (2)-(9) represent should be included; otherwise, it isn't easy to follow the logic of the authors' considerations, or it is necessary to access the bibliographic references used. The observation is also valid for equations (11) and (12), introduced in the paper, without explaining how they arrived.

9. Is there a justification for using exponential functions when modeling hardness variation? Were several functions tested, and did the exponential function best fit the experimental results?

10. On page 5, lines 139-140, the authors show that "The higher the applied pressure, the more compact powder particles and the more metallic bonds are created to resist the deformation, reflected in the  higher value in the hardness test". I believe that the higher hardness values are obtained later due to the sintering process applied after compaction.

9. The software used for the mathematical processing of the experimental results and the software used for developing the diagrams should be mentioned.

10. The brand of the analytical balance used to evaluate the masses, the brand of the equipment on which the powder was pressed (how was the force or pressure measurement performed?), etc., could also be mentioned.

11. Authors must pay more attention to article editing and English expression. The authors could ask a person with ample knowledge of English to check the text, as some wording is unclear. For example, " During compaction, Mg powder successively has rearrangement, elastic deformation, and initial and severe plastic deformation [68].”, ”The lowest value appears in the middle part, the distance h from the top.”, ”The hardness decreases E exponentially ...”,In this study, the discussion about the relationship between applied pressure and hardness distribution is valuable in predicting and improving sample quality" (valuable?), etc.

The texts in some of the figures (for example, those in figures 2, 4a, 4b, etc.) are written in too small fonts.

Explaining symbols immediately after an equation must be done using the word "where". Explanations of the symbols used should be inserted after the equations (1)-(6), etc.

In the case of bibliographic reference no. 4, words in the title of the paper/work cited could start with a lowercase letter, except for the first word. In the case of reference no. 4, the last name of the third author is misspelled (mayer instead of Mayer). It is unclear to me whether the paper appeared in the mentioned journal (Addit. Manuf, 2017, 19, 4); on the web (https://onlinelibrary.wiley.com/action/doSearch?ContribAuthorRaw=Danninger%2C+Herbert, or https://publik.tuwien.ac.at/files/publik_263735.pdf), the work appears to be an article from an encyclopedia (Ullmann's Encyclopedia of Industrial Chemistry).

According to the recommendations of the web page https://www.mdpi.com/journal/jmmp/instructions, in the list of bibliographic references, a comma should not be placed after the names of the authors.

Author Response

(The authors gave the same response as above.)
